# Maximum a Posteriori on a Submanifold: a General Image Restoration Method with GAN

## Abstract

We propose a general method for various image restoration problems, such as denoising, deblurring, super-resolution and inpainting. The problem is formulated as a constrained optimization problem. Its objective is to maximize a posteriori probability of latent variables, and its constraint is that the image generated by these latent variables must be the same as the degraded image. We use a Generative Adversarial Network (GAN) as our density estimation model. Convincing results are obtained on MNIST dataset.

## 1 Introduction

Image restoration has been researched for many years, but in a case-by-case way (Park et al., 2003; Mairal et al., 2008; Guillemot & Le Meur, 2014). Almost all image restoration algorithms are only designed for certain type of images or degradation. This research paradigm has some obvious disadvantages. It is exhausting to invent new algorithms or train new models for slightly different situations. Even if we can, those specialized solutions are not so elegant, because they are very unlike one another even though the problems they focus on are fundamentally so similar.

It is worth noting that any image degradation process can be abstracted as a many-to-one function. More specifically, for any given degradation process, one degraded image could be degraded from many possible original images. From that point of view, we propose a general method for various image restoration problems, such as denoising, deblurring, super-resolution and inpainting. Our algorithm chooses the most probable original image from all those possible original images, and uses it as the restoration of the given degraded image. To be more precise, the general image restoration is formulated as a constrained optimization problem. Its objective is to maximize a posteriori probability of latent variables, and its constraint is that the image generated by these latent variables must be the same as the degraded image.

Recent progress of density estimation techniques makes our algorithm possible. In the field of image generation, Generative Adversarial Networks (GANs) make a huge success in recent years (Goodfellow et al., 2014; Radford et al., 2015). As research continues, images generated by GANs become more and more realistic and clear, and training procedure of GANs become more and more stable (Salimans et al., 2016; Arjovsky et al., 2017). Besides being an image generation technique, GANs can also be used for density estimation. The generator part of a GAN is an implicit probability distribution model, and it will converge to a good estimator of the data distribution after training. In this work, we solve the inference problem with the probability density estimated by a GAN.

Figure 1 provides an illustration of how our image restoration method works. There are four dashed boxes from left to right in Figure 1, corresponding to four different phases of image capture and restoration process. Images in the first dashed box are original images, which are clear and undegraded. These images undergo a series of degradation in the second dashed box, and then are captured by our camera. In the image restoration process, we hope to estimate the original images with the degraded images we captured. As we pointed out before, every degraded image could be degraded from many possible original images. To be more precise, there is a particular subset of the original image manifold for any degraded image, and all image samples on the submanifold could be degraded to the given degraded image. Images in the third dashed box are those samples on the submanifold, and they are arranged in ascending order of log-likelihood from left to right. Images marked by yellow boxes are samples with the highest log-likelihood in their group, and they are placed in the last dashed box as restoration outputs.

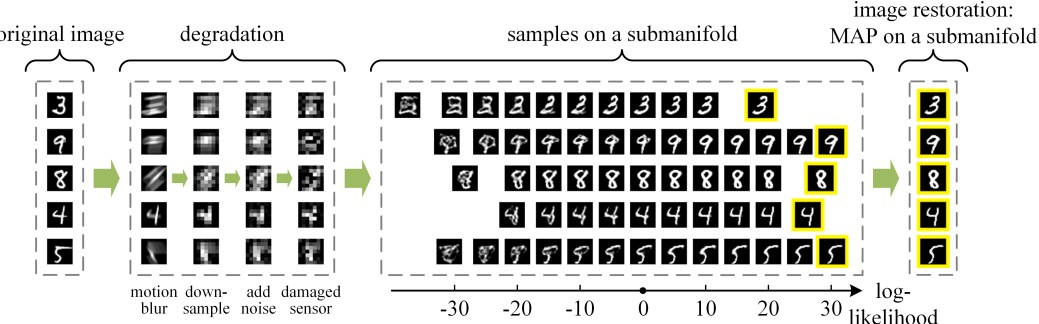

Figure 1: An illustration of our image restoration method.

Overall, the contributions of this work are mainly in two aspects:

1. We propose a general method for various image restoration problems. In the method, we explicitly use density information estimated by a GAN, an implicit model; and we directly solve the image restoration problem, an inference problem, with a GAN, a generative model. To the best of our knowledge, our work is the first to do those two things.

2. We propose a new algorithm to solve the optimization problem in our method. The new optimization algorithm is a first-order iterative algorithm for constrained problems, and it works well even for problems with highly nonlinear objective and constraints. These features make it especially suited to neural network related constrained optimization problems.

## 2  RELATED WORK

The most similar work to ours is proposed in Yeh et al. (2017). They propose a semantic image inpainting method, which can generate missing content with a trained GAN. They search in the latent space of the trained GAN for the image which is closest to the corrupted image, and use the discriminator loss of the trained GAN as an indicator of how realistic their restoration is. Their motivation is similar to our work, but unfortunately, there is a major theoretical flaw in their method. Goodfellow et al. (2014) already prove that the discriminator is unable to identify how realistic an input is after several steps of training, if the GAN has enough capacity. During the training, the information of the data distribution gradually transfer from the discriminator to the generator. Ideally, the generator will have all the information of the data distribution while the discriminator will have none. That is why we use the generator of a trained GAN as an implicit probability density model in our method. Another difference between their work and ours is that they only focus on image inpainting problem, while our method applies to various image restoration problems.

The maximum a posteriori (MAP) has existed for a long time as a classic estimation method (Campisi & Egiazarian, 2016). But before GANs, people do not have a probability density model which is good enough to describe the distribution of images. After GANs make a huge success in image generation, researchers start to use them in image restoration tasks to get more realistic results (Isola et al., 2017; Bousmalis et al., 2017). Ledig et al. (2017) and Sønderby et al. (2016) try to use the MAP estimation on GANs to solve image super-resolution problem. However, they only use the MAP estimation implicitly and indirectly, while our method use it explicitly and directly. We suspect that all methods do implicit MAP estimation on GANs would require redesigning or retraining when the image restoration task changes, and this makes implicit methods not as general as our explicit method.

Ulyanov et al. (2017) is another work which is seemingly similar to ours, but they are actually quite different. Their work uses a randomly-initialized neural network as an image prior to solve various image restoration problems. The prior in their method is elaborate, neural network related but still handcrafted, while in our method the prior is learned from data. So our data-driven prior has better adaptability to specific image distribution.

## 3 MAXIMUM A POSTERIORI ON A SUBMANIFOLD

### 3.1 FORMULATION

Consider a general image degradation model as follows,

$$\tilde{x} = F(x, \Omega) \tag{1}$$

where $x$, $\tilde{x}$, and $\Omega$ represent the original image, the degraded image, and the parameters of the degradation model, respectively. The image degradation function $F$ is a deterministic function. That means, given an original image $x$ and a particular set of parameters $\Omega$, the image degradation model will always produce the same degraded image $\tilde{x}$.

Our goal is to get a reasonable estimate of $x$ with given $\tilde{x}$ and $F$. In this paper, we use the maximum a posteriori probability (MAP) estimate of $x$ as the restoration of $\tilde{x}$. Compared to MSE-based method, MAP estimate of $x$ is perceptually more convincing. We can perform inference by maximizing the posterior $p(x, \Omega|\tilde{x})$:

$$\begin{aligned}
\{\hat{x}, \hat{\Omega}\} &= \underset{x, \Omega}{\arg\max}\, p(x, \Omega|\tilde{x}) \\
&= \underset{x, \Omega}{\arg\max}\, \frac{p(\tilde{x}|x, \Omega)p(x|\Omega)p(\Omega)}{p(\tilde{x})}
\end{aligned} \tag{2}$$

where $\hat{x}$ and $\hat{\Omega}$ represent MAP estimate of $x$ and $\Omega$. Note that $p(\tilde{x})$ is always positive and does not depend on $x$ and $\Omega$, and typically we assume that $x$ and $\Omega$ are independent. Therefore,

$$\{\hat{x}, \hat{\Omega}\} = \underset{x, \Omega}{\arg\max}\, p(\tilde{x}|x, \Omega)p(x)p(\Omega) \tag{3}$$

Note that $\tilde{x} = F(x, \Omega)$ is a deterministic function, i.e., $p(\tilde{x}|x, \Omega) = \delta(\tilde{x} - F(x, \Omega))$. Therefore, the estimation is equivalent to

$$\begin{aligned}
\{\hat{x}, \hat{\Omega}\} = \underset{x, \Omega}{\arg\max}\quad & p(x)p(\Omega) \\
\text{s.t.}\quad & \|\tilde{x} - F(x, \Omega)\| = 0
\end{aligned} \tag{4}$$

Here we write $p(x)$ more specifically as $p_r(x)$, which stand for the probability density of real data distribution. We can estimate $p_r(x)$ with the generator part of a trained GAN, which is an implicit probability distribution model with distribution $p_G(x)$. The trained generator $G$ represents a mapping from latent space of $z$ to data distribution of original image $x$, i.e., $p_r(x) = p_G(x)$, and $p_G(x)$ is a probability density function implicitly defined by $x = G(z)$, where $z$ is typically sampled from some simple distribution, such as the uniform distribution or the normal distribution. Assuming $G : \mathbf{R}^n \to \mathbf{R}^m$ is an injective function, the estimation is equivalent to

$$\begin{aligned}
\{\hat{z}, \hat{\Omega}\} = \underset{z, \Omega}{\arg\max}\quad & p_G(G(z))p(\Omega) \\
\text{s.t.}\quad & \|\tilde{x} - F(G(z), \Omega)\| = 0
\end{aligned} \tag{5}$$

$$\text{and}\quad \hat{x} = G(\hat{z}) \tag{6}$$

Generally the dimension of vector space of $z$ is far lower than the dimension of vector space of $x$. Note that $p_G(x)$ is nonnegative if and only if $x$ is on the low dimensional manifold $\mathcal{M}$ defined by $x = G(z)$, we can replace the probability density on the original space $p_G(G(z))$ in Eq. (5) by the probability density on the manifold $p_{\mathcal{M}}(z)$, and end up with the same estimation result $\hat{z}$. According to Pennec (2004), the probability density on the manifold can be calculated by

$$p_{\mathcal{M}}(z) = \frac{p(z)}{\sqrt{\det Gram(\frac{\partial G}{\partial z_1}, \dots, \frac{\partial G}{\partial z_n})}} \tag{7}$$

where $Gram$ represents the Gram matrix, and $\sqrt{\det Gram(\frac{\partial G}{\partial z_1}, \dots, \frac{\partial G}{\partial z_n})}$ is the volume of the parallelotope spanned by the vectors $(\frac{\partial G}{\partial z_1}, \dots, \frac{\partial G}{\partial z_n})$, so the square root of the Gram determinant can serve as a local scale factor. It has an effect similar to the Jacobian determinant, but we can only

use the Gram determinant here because $G$ is a function from $\mathbf{R}^n$ to $\mathbf{R}^m$, and generally $n$ is much less than $m$.

The Gram matrix can be simply calculated by $Gram(\frac{\partial G}{\partial \boldsymbol{z}_1}, \dots, \frac{\partial G}{\partial \boldsymbol{z}_n}) = \boldsymbol{V}^T \boldsymbol{V}$, where $\boldsymbol{V}$ is an $m \times n$ matrix, whose entries are given by $\boldsymbol{V}_{ij} = \frac{\partial \boldsymbol{x}_i}{\partial \boldsymbol{z}_j}$. Therefore, Eq. (5) is equivalent to

$$\{\hat{\boldsymbol{z}}, \hat{\boldsymbol{\Omega}}\} = \underset{\boldsymbol{z}, \boldsymbol{\Omega}}{\operatorname{argmax}} \quad \frac{p(\boldsymbol{z})p(\boldsymbol{\Omega})}{\sqrt{\det \boldsymbol{V}^T \boldsymbol{V}}} \tag{8}$$
$$\text{s.t.} \quad \|\tilde{\boldsymbol{x}} - F(G(\boldsymbol{z}), \boldsymbol{\Omega})\| = 0$$

To solve the estimation problem efficiently, we represent probabilities in Eq. (8) in logarithmic space, i.e.,

$$\{\hat{\boldsymbol{z}}, \hat{\boldsymbol{\Omega}}\} = \underset{\boldsymbol{z}, \boldsymbol{\Omega}}{\operatorname{argmax}} \quad -\frac{1}{2} \log \det \boldsymbol{V}^T \boldsymbol{V} + \log p(\boldsymbol{z}) + \log p(\boldsymbol{\Omega}) \tag{9}$$
$$\text{s.t.} \quad \|\tilde{\boldsymbol{x}} - F(G(\boldsymbol{z}), \boldsymbol{\Omega})\| = 0$$

Matrix $\boldsymbol{V}^T \boldsymbol{V}$ is a positive-definite matrix, so we can use Cholesky decomposition to calculate $\log \det \boldsymbol{V}^T \boldsymbol{V}$ efficiently, i.e.,

$$\log \det \boldsymbol{V}^T \boldsymbol{V} = 2 \operatorname{tr}(\log(\operatorname{chol}(\boldsymbol{V}^T \boldsymbol{V}))) \tag{10}$$

Finally we deduce a set of expressions which can be calculated directly, and their final outcome $\hat{\boldsymbol{x}}$ is the restored image we want, i.e.,

$$\{\hat{\boldsymbol{z}}, \hat{\boldsymbol{\Omega}}\} = \underset{\boldsymbol{z}, \boldsymbol{\Omega}}{\operatorname{argmax}} \quad -\operatorname{tr}(\log(\operatorname{chol}(\boldsymbol{V}^T \boldsymbol{V}))) + \log p(\boldsymbol{z}) + \log p(\boldsymbol{\Omega}) \tag{11}$$
$$\text{s.t.} \quad \|\tilde{\boldsymbol{x}} - F(G(\boldsymbol{z}), \boldsymbol{\Omega})\| = 0$$
$$\text{and} \quad \hat{\boldsymbol{x}} = G(\hat{\boldsymbol{z}}) \tag{12}$$

Note that $(G(\boldsymbol{z}), \boldsymbol{\Omega})$ form a low dimensional manifold which is embedded in the space of $(\boldsymbol{x}, \boldsymbol{\Omega})$, and the feasible solutions of Eq. (11) is on a subset of the manifold, which is defined by $\|\tilde{\boldsymbol{x}} - F(G(\boldsymbol{z}), \boldsymbol{\Omega})\| = 0$. So our method basically makes a MAP estimate on a submanifold.

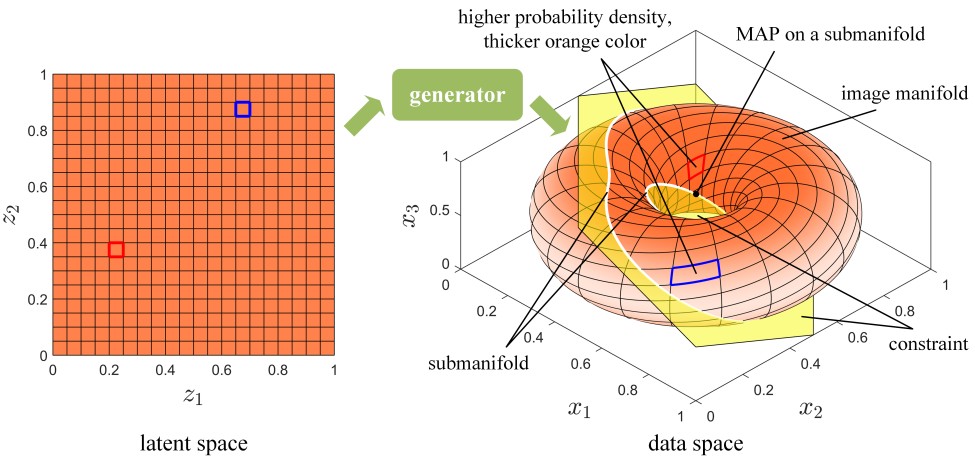

Figure 2: A toy example to show how our image restoration method works.

Figure 2 is a toy example to show how our method works in a very visible way. Suppose there is a grayscale original image $\boldsymbol{x}$, which has only three pixels. Then it is downsampled to only one pixel during the image capture process, and our task is to estimate $\boldsymbol{x}$ with the one pixel image we captured. Suppose we have trained a GAN as an implicit model of data distribution of $\boldsymbol{x}$. More specifically, the generator of the trained GAN represents a mapping from its input noise $\boldsymbol{z}$ to data distribution of $\boldsymbol{x}$. The left part of Figure 2 describes the two dimensional latent space of $\boldsymbol{z}$. We use the saturation of orange color to represent probability density level, i.e., a thicker orange color means a higher

probability density. So the uniform orange color in the latent space means that the input noise $z$ is sampled from a uniform distribution.

Then the two dimensional vector $z$ is mapped to three dimensional space of image $x$ by the generator of the trained GAN, and the big orange square in the latent space of $z$ is transformed into a twisted torus in the three dimensional data space of $x$, which is described in the right part of Figure 2. Some areas in space of $z$ expand during the transformation, while other areas shrink. We can find this out by comparing the red and blue quadrilateral between the latent and data space. Therefore, the probability density on the torus is no longer uniform. The orange colors of the expanded areas become lighter, and the colors of the shrunken areas become thicker. Quantitatively speaking, the square root of the Gram determinant in Eq. (7) is the local area scale factor of the mapping, and its inverse, of course, is the local density scale factor.

The pale yellow plane in the data space represents the constraint in the toy example. All points on the plane would exactly be downsampled to the one pixel image we captured. So the intersection curve of the plane and the torus is the submanifold we are looking for, and that white curve is the feasible set of the toy problem. In this problem, $p(z)$ is a constant in the domain, and degradation parameters $\Omega$ does not exist at all. According to Eq. (8), what we need to do is to maximize the inverse of the square root of the Gram determinant on the submanifold. In other words, the point with the thickest orange color on the intersection curve is the restored image $\hat{x}$, the MAP estimate on the submanifold. We can find out that the method is both intuitive and rational for this toy example.

## 3.2 Optimization Algorithm

We propose a new optimization algorithm to solve Eq. (11). Note that the objective function and the equality constraint in Eq. (11) are both highly nonlinear, so gradient-based method seems a natural choice for the problem. Our algorithm is inspired by Projected Gradient Descent Method.

To solve a unconstrained problem with ordinary Gradient Descent Method, we take small steps in the direction of the negative gradient. To solve an constrained problem, we can try to use Projected Gradient Descent Method, take small step as usual and then project variables back onto the feasible set. But unfortunately, Projected Gradient Descent Method is only valid for problems with very simple feasible set, such as solution set of linear equations, some simple polyhedra and simple cone, etc. If constraints of a problem is too complex, like the constraint in Eq. (11), it is very hard to project variables back onto the feasible set.

To overcome this shortage, we propose a new optimization algorithm called Quasi Projected Gradient Descent Method. In our algorithm, the gradient information is not only used to improve the objective function, but helps to satisfy the constraints as well. Consider the standard form of continuous optimization problem,

$$
\begin{aligned}
\underset{u}{\text{minimize}} \quad & f(u) \\
\text{s.t.} \quad & h_i(u) = 0, \ \ i = 1, \dots, m \\
& h_j(u) \leq 0, \ \ j = m+1, \dots, m+p
\end{aligned} \tag{13}
$$

where $f, h_i, h_j : \mathbf{R}^n \to \mathbf{R}$, and they are all highly nonlinear. Algorithm 1 is the proposed algorithm for the problem.

To solve Eq. (11) using the proposed algorithm, we only need to set $u = \{\hat{z}, \hat{\Omega}\}$, objective function $f(u) = -(-\operatorname{tr}(\log(\operatorname{chol}(V^T V))) + \log p(z) + \log p(\Omega))$, and the only equality constraint function $h_1(u) = \|\tilde{x} - F(G(z), \Omega)\|$.

In the proposed algorithm, we first define an overall constraint function $h(u) : \mathbf{R}^n \to \mathbf{R}_{\geq 0}$, and the feasible set of the optimization problem is the region where $h(u) = 0$. In each iteration of the algorithm, we calculate the gradients of $f(u)$ and $h(u)$ at $u_{i-1}$. If we take a small step in the direction of the negative $g_f$, the value of $f(u)$ will decrease a little bit, but it may have a unwanted impact on the value of $h(u)$. In order to avoid this problem, we calculate $g_\parallel$, the tangential component of $g_f$ on the isocontour of $h(u_{i-1})$, which can be calculated by vector rejection of $g_f$ on $g_h$. In each iteration, we actually take a small step in the direction of the negative $g_\parallel$, the value of $f(u)$ will still decrease, while it has almost no impact on the value of $h(u)$. We also take a small step in the direction of the negative $g_\perp$, i.e., $g_h$ itself, which is perpendicular to the isocontour of

---

**Algorithm 1** Quasi Projected Gradient Descent Method

---

**Require:** step size $\eta_\parallel$ and $\eta_\perp$, positive factors $c_i$ and $c_j$, number of iterations $n$, small positive constant $\epsilon$ for numerical stability, initial guess $\boldsymbol{u}_0$

Define $h(\boldsymbol{u}) = \sum\limits_{i=1}^{m} c_i \cdot \|h_i(\boldsymbol{u})\|^2 + \sum\limits_{j=m+1}^{m+p} c_j \cdot H(h_j(\boldsymbol{u})) \cdot \|h_j(\boldsymbol{u})\|^2$, where $H$ represents the Heaviside step function

**for** $i = 1$ **to** $n$ **do**

    $\boldsymbol{g}_f = \nabla f(\boldsymbol{u}_{i-1})$

    $\boldsymbol{g}_h = \nabla h(\boldsymbol{u}_{i-1})$

    $\boldsymbol{g}_\parallel = \boldsymbol{g}_f - \frac{\boldsymbol{g}_f \cdot \boldsymbol{g}_h}{\boldsymbol{g}_h \cdot \boldsymbol{g}_h + \epsilon} \cdot \boldsymbol{g}_h$

    $\boldsymbol{g}_\perp = \boldsymbol{g}_h$

    $\boldsymbol{u}_i = \boldsymbol{u}_{i-1} - \eta_\parallel \cdot \boldsymbol{g}_\parallel$ (or do with an advanced gradient descent optimizer)

    $\boldsymbol{u}_i = \boldsymbol{u}_i - \eta_\perp \cdot \boldsymbol{g}_\perp$ (or do with another advanced gradient descent optimizer)

**end for**

**return** $\boldsymbol{u}_n$ and $f(\boldsymbol{u}_n)$

---

$h(\boldsymbol{u}_{i-1})$. Repeat these steps, and the sequence $\boldsymbol{u}$ will hopefully converge to the desired optimal solution.

Behaviors of our Quasi Projected Gradient Descent Method is similar to behaviors of the original Projected Gradient Descent Method. Consider a point $\boldsymbol{u}$ which is very close to the feasible region. The summation of two moves against $\boldsymbol{g}_\parallel$ and $\boldsymbol{g}_\perp$ is actually an inaccurate Projected Gradient Descent. That is why we name our method as "Quasi Projected Gradient Descent Method".

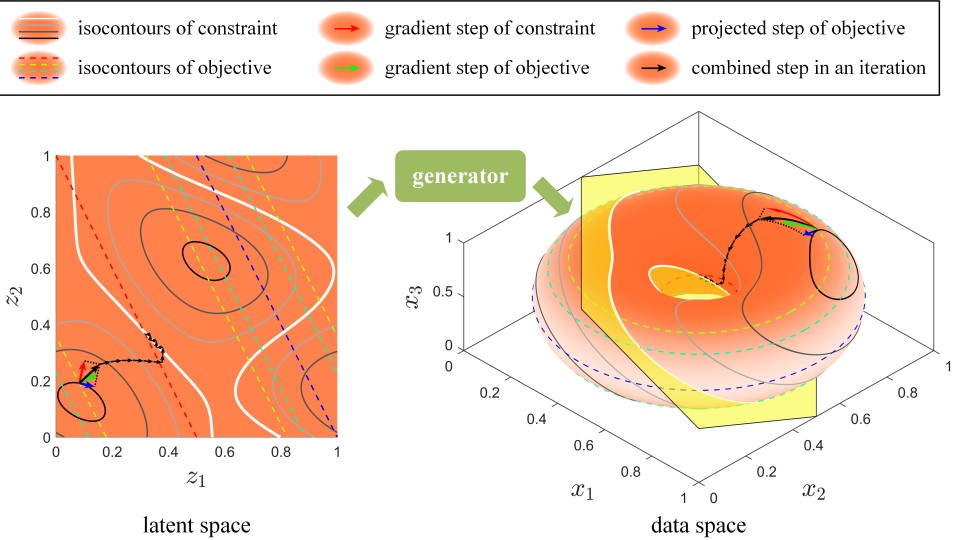

Figure 3: A toy example to show how our Quasi Projected Gradient Descent Method works.

Here we use the same toy example we used in Section 3.1, to show how our Quasi Projected Gradient Descent Method works. In Figure 3, solid curves in black and white are isocontour of constraint function $h$. The whiter the curve, the lower value of $h$ it corresponds; Dashed lines in color are isocontour of objective function $f$. The redder the line, the lower value of $f$ it corresponds. Note that the white solid curve is the feasible set of the toy problem, so intersection points of the white solid curve and the red dashed line in the latent space is $\hat{\boldsymbol{z}}$ in Eq. (11), while the intersection points in the data space is $\hat{\boldsymbol{x}}$ in Eq. (12).

Our iterative optimization algorithm starts from the bottom left corner of the latent space. The red vector is a gradient step of $h$. It is pointing towards the direction of the negative $\boldsymbol{g}_h$, and is perpendicular to the black solid curve, an isocontour of $h$. The green vector is a gradient step of $f$. It is pointing towards the direction of the negative $\boldsymbol{g}_f$, and is perpendicular to the yellow dashed

line, an isocontour of $f$. The blue vector is a projected gradient step. It is pointing towards the direction of the negative $g_{\parallel}$, and is the tangential component of the green vector on the black solid curve, which can be calculated by vector rejection of the green vector on the red vector. We only plot green, red and blue vector for the first iteration to keep Figure 3 clean and easy to understand. Black vectors are combined gradient steps, which are vector sums of red and blue vectors. We move along these black vectors and we can find out that our optimization algorithm reaches a desired solution quickly.

## 4 EXPERIMENTS

In this Section, we use MNIST dataset (LeCun et al., 1998) to test our image restoration method. The dataset is divided in 50k for the training set, 10k for each of the validation and test set. We use a WGAN-GP (Gulrajani et al., 2017) trained on the training set as the density estimation model. The architecture of the WGAN-GP we used is shown in Table 1 and Table 2, and we add a L2 weight decay term with decay parameter of 0.001 on the generator loss to prevent over-fitting. The network we used is very simple, but it is enough to prove the effectiveness of our method.

Table 1: Architecture of the generator

|  | Kernel size | Output shape |
|---|---|---|
| $z$ | - | 16 |
| Linear, tanh | - | $64 \times 4 \times 4$ |
| Deconv, tanh | $5 \times 5$ | $32 \times 7 \times 7$ |
| Deconv, tanh | $5 \times 5$ | $16 \times 14 \times 14$ |
| Deconv, sigmoid | $5 \times 5$ | $1 \times 28 \times 28$ |

Table 2: Architecture of the discriminator

|  | Kernel size | Output shape |
|---|---|---|
| $G(z)$ | - | $1 \times 28 \times 28$ |
| Conv, LeakyReLU | $5 \times 5$ | $16 \times 14 \times 14$ |
| Conv, LeakyReLU | $5 \times 5$ | $32 \times 7 \times 7$ |
| Conv, LeakyReLU | $5 \times 5$ | $64 \times 4 \times 4$ |
| Linear | - | 1 |

We use four different kinds of degradation to test the generality of our method. The first three kinds of degradation are relatively simple. They are $7\times$ downsampling, making a $14\times14$ square hole in the center of the image, and adding Gaussian white noise with a standard deviation of 1.0, respectively. The last kind of degradation is a composition of a series of degradation in order, which are (a) adding linear motion blur by at most 14 pixels in any direction, (b) $4\times$ downsampling, (c) adding uniform noise between -0.05 and 0.05, (d) randomly removing 10% of the pixels.

We use two independent ADAM optimizer (Kingma & Ba, 2014) with $g_{\parallel}$ and $g_{\perp}$ respectively in the Quasi Projected Gradient Descent Method. For all four kinds of degradation, we run the algorithm with the same settings. Settings for both ADAM optimizer are learning rate $\alpha = 0.01$ (decayed linearly to 0), $\beta_1 = 0.9$, $\beta_2 = 0.99$, and number of iterations $n = 500$.

To the best of our knowledge, there is only one other general image restoration algorithm which can cope with various kinds of degradation like ours does. And that is the nearest neighbor algorithm. It searches in the training set for an image, whose degradation is the nearest to the given degraded image $\tilde{x}$. More specifically, if there are $m$ points $x^{(1)}, \ldots, x^{(m)}$ in the training set, then

$$\{\hat{x}_{NN}, \hat{\Omega}_{NN}\} = \underset{x^{(i)}, \Omega}{\operatorname{argmin}} \|\tilde{x} - F(x^{(i)}, \Omega)\| \tag{14}$$

where $\hat{x}_{NN}$ represents the restored image by the nearest neighbor algorithm. In case of multiple occurrences of the minimum objective values, we choose the one with the largest $p(\hat{\Omega}_{NN})$. Note that the empirical distribution of the training set is

$$p_e(x) = \frac{1}{m} \sum_{i=1}^{m} \delta(x - x^{(i)}) \tag{15}$$

If we replace $p(x)$ with $p_e(x)$ rather than $p_G(x)$ in Eq. (4), or in other words, if the generative model in our method is extremely over-fitting, our method will degenerate to the nearest neighbor algorithm. So our method can be treated as a generalized method of the nearest neighbor algorithm. In the experiments, we use the nearest neighbor algorithm as a baseline, and compare its results with the proposed method.

Table 3: Visual results

|  | **Downsample** | **Hole** |
|---|---|---|
| Original image |  |  |
| Degraded image | | |
| Nearest neighbor | | |
| Our restoration | | |

|  | **Noise** | **Composition** |
|---|---|---|
| Original image |  |  |
| Degraded image | | |
| Nearest neighbor | | |
| Our restoration | | |

Table 4: Quantitative results

|  |  | **Downsample** | **Hole** | **Noise** | **Composition** |
|---|---|---|---|---|---|
| $MSE(\tilde{\boldsymbol{x}}, F(\hat{\boldsymbol{x}}, \hat{\boldsymbol{\Omega}}))$ | NN | 1.0e-3 | 0.0061 | 0 | 0.0025 |
|  | Ours | 7.2e-5 | 0.0014 | 0 | 6.4e-5 |
| $MSE(\boldsymbol{x}, \hat{\boldsymbol{x}})$ | NN | 0.043 | 0.037 | 0.039 | 0.056 |
|  | Ours | 0.026 | 0.034 | 0.034 | 0.043 |
| $MSSSIM(\boldsymbol{x}, \hat{\boldsymbol{x}})$ | NN | 0.73 | 0.76 | 0.77 | 0.69 |
|  | Ours | 0.84 | 0.77 | 0.80 | 0.76 |

Experimental results are shown in Table 3 and Table 4. $MSE(\tilde{\boldsymbol{x}}, F(\hat{\boldsymbol{x}}, \hat{\boldsymbol{\Omega}}))$ is a measure of how accurately a restored image can be degraded back to its input, and $MSE(\boldsymbol{x}, \hat{\boldsymbol{x}})$ and $MSSSIM(\boldsymbol{x}, \hat{\boldsymbol{x}})$ (Wang et al., 2003) are measures of the difference between the restoration and the ground truth. We can find out that our general image restoration method is better than the baseline method. The nearest neighbor algorithm cannot use the information of the probability density of images well, and that is the major disadvantage compared to our method.

## 5 CONCLUSIONS AND FUTURE WORK

We propose a general image restoration method in this work. Compared with traditional image restoration algorithms, our method is much more powerful. Image restoration is an inherently ill-posed problem, so additional prior knowledge is needed. In our method, we use all prior knowledge of original images, i.e., the probability distribution of original images; and we use all prior knowledge of degradation, i.e., the degradation model itself. Traditional image restoration, by contrast, just uses a small part of the prior, typically some statistical properties. Besides, unlike our method, there is usually no guarantee that an output restoration from a traditional method can be degraded back accurately to its input. This makes restorations from a traditional method less plausible than restorations from our method.

For future work, We think our method can be straightforwardly extended to other domains which GANs are gifted in, such as video, audio and language. We will try to solve restoration problems and other inference problems in these domains with our paradigm. The convergence and other properties of the Quasi Projected Gradient Descent Method would be interesting as well.

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
