# OpenReview forum: "Maximum a Posteriori on a Submanifold: a General Image Restoration Method with GAN"
_ICLR.cc/2019/Conference_

### Official Review · AnonReviewer2 · 2018-10-29
**Interesting idea, but the paper needs improvements.**

**Rating:** 6
**Confidence:** 4

**Review:**

The authors propose a method for image restoration, where the restored image is the MAP estimate. A pretrained GAN is utilized to approximate the prior distribution of the noise-free images. Then, the likelihood induces a constraint which is based on the degradation function. In particular, the method tries to find the latent point for which the GAN generates the image, which if gets degraded will match the given degraded image. Also, an optimization algorithm is presented that solves the proposed constrained optimization problem.

I find the paper very well written and easy to follow. Also, the idea is pretty clean, and the derivations are simple and clear. Additionally, the Figures 2,3 are very intuitive and nicely explain the theory. However, I think that there are some weaknesses (see comments):

Comments:

#1) I do not understand exactly what the "general method" means. Does it mean that you propose a method, where you can just change the F, such that to solve a different degradation problem? So you provide the general framework where somebody has to specify only the F?

#2) Clearly, the efficiency of the method is highly based on the ability of the GAN to approximate well the prior distribution of the noise-free images.

#3) There are several Equations that can be combined, such that to save enough white space in order to discuss further some actual technical details. For instance, Eq. 2,3 can be easily combined using the proportional symbol, Eq. 8,9,10,11 show actually the same thing.

#4) I think that the function F has to be differentiable, and this should be mentioned in the text. Also, I believe that some actual (analytic) examples of F should be provided, at least  in the experiments. The same holds for the p(Omega). This parameter Omega is estimated individually for each degraded image?

#5) Before Eq. 8 the matrix V is a function of z and should be presented as such in the equations.

#6) I believe that it would be nice to include a magnified image of Fig. 3, where the gradient steps are shown. Also, my understanding is that the optimization goal is to find first a feasible solution, and then find the point that maximizes f. I think that this can be clarified in the text.

#7) The optimization steps seem to be intuitive, however, there is not any actual proof of converge. Of course, the example in the Figure 3 is very nice and intuitive, but it is also rather simple. I would suggest, at least, to include some empirical evidences in the experiments that show convergence.

#8) In the experiments I think that at least one example of F and p(Omega) should be presented. Also, what the numbers in Table 4 show? Which is the best value that can be achieved? These numbers correspond to several images, or to a unique image?
#9) I think that MNIST is almost a toy experiment, since the crucial component of the proposed method is the prior modeling with the GAN. I believe that a more challenging experiment should be conducted e.g. using celebA dataset.

Minor comments:

#1) In the paragraph after Eq. 4 the equality p_r(x)=p_G(x) is very strong assumption. I would suggest to use the \simeq symbol instead.

#2) After Eq. 6 the "nonnegative" should be "nonzero".

#3) Additional density estimation models can be used e.g. VAEs, GMM. Especially, I believe that the VAE will provide a way to approximate the prior easier than the GAN.

#4) In Section 2 paragraph 2, the sentence "However, they only ... and directly" is not clear what means.

In general, I find both the proposed model and optimization algorithm interesting. Additionally, the idea is nicely presented in the paper. Most of my comments are improvements which can be easily included. The two things that make me more skeptical, is the convergence of the proposed algorithm and the experiments. The MNIST is a relatively simple experiment, and I would like to see how the method works in more challenging problems. Also, I think that additional methods to compute the image prior should be included in the experiments.

---

> ### Author Response · Authors · 2018-11-27
> **Response to AnonReviewer2**
>
> Thank you for your thoughtful review. We will address your concerns in turn.
>
> Q1: So you provide the general framework where somebody has to specify only the F?
> A1: Yes, and that is the motivation of this work, to avoid training new models for slightly different situations.
>
> Q2: The efficiency of the method is highly based on the ability of the GAN to approximate well the prior distribution of the noise-free images.
> A2: Yes, so we use WGAN-GP, a strong and elegant implementation, as our trained GAN.
>
> Q3: Is parameter Omega estimated individually for each degraded image?
> A3: Yes.
>
> Thank you again for your positive reviews which give me some confidence, I really appreciate it.

---

### Official Review · AnonReviewer1 · 2018-10-30
**Nice presentation but missing important reference**

**Rating:** 4
**Confidence:** 5

**Review:**

This paper proposed a general method for image restoration based on GAN. In particular, the latent variable z is optimized based on the MAP framework. And the results are obtained by G(z). This method looks reasonable to achieve good results. However, the idea is very related to Yeh et al.’s work which has already published but not mentioned at all.

Yeh, Raymond A., et al. "Image Restoration with Deep Generative Models." 2018 IEEE International Conference on Acoustics, Speech and Signal Processing (ICASSP). IEEE, 2018.

Both the proposed method and Yeh et al.’s method optimize the latent variable z of the generator using MAP, although the loss functions are slightly different. In addition, the applications are very similar: image inpainting, denoising, super-resolution etc. Yeh et al.’s method should be the right baseline instead of the nearest neighbor algorithm.

In addition, the results seem very weak. There are tons of algorithms for image inpainting, denoising, and super-resolution, but the proposed method was not compared with them. The paper claims that only the nearest neighbor algorithm can handle different degradations. This is not true. For example, total variation regularization can do all these tasks.

Some other comments: what are the parameters of the degradation in the applications? For example, in image inpainting, does the proposed method learn the mask as well? So it is blind inpainting?

---

> ### Author Response · Authors · 2018-11-27
> **Response to AnonReviewer1**
>
> Thank you for your thoughtful review. We will address your concerns in turn.
>
> Q1: The idea is very related to Yeh et al.’s work which is not mentioned at all.
>
> A1: The entire first paragraph of our related work section is focused on Yeh et al.’s work. As we explained in the paragraph, there is a major theoretical flaw in their method. Yeh et al. (2017) use the discriminator loss of a trained GAN as an indicator of how realistic their restoration is. However, Goodfellow et al. (2014) already prove that the discriminator is unable to identify how realistic an input is after several steps of training, if the GAN has enough capacity. Ideally the generator will have all the information of the data distribution while the discriminator will have none. That is why we use the generator of a trained GAN as an implicit probability density model in our method.
>
> Another difference between their work and ours is that they only focus on image inpainting problem, while our method applies to various image restoration problems.
>
>
> Q2: Total variation regularization can also handle different degradations.
>
> A2: We think you underestimate the difficulty of those restoration problems. Please check the degraded images in Table 3. These images are damaged so badly that TV cannot recover any meaningful thing. As a handcrafted prior, TV performs much worse than our data-driven baseline method in these tasks.
>
>
> Q3: Does the proposed method learn the image inpainting mask as well? What are the parameters of the degradation in the applications?
>
> A3: The image inpainting mask is known and fixed.
>
> We use four different kinds of degradation to test the generality of our method. The first three kinds of degradation are 7× downsampling, making a 14×14 square hole in the center of the image, and adding Gaussian white noise with a standard deviation of 1.0, respectively. The last kind of degradation is a composition of a series of degradation in order, which are (a) adding linear motion blur by at most 14 pixels in any direction, (b) 4× downsampling, (c) adding uniform noise between -0.05 and 0.05, (d) randomly removing 10% of the pixels.

---

> > ### Comment · AnonReviewer1 · 2018-11-27
> > **Comments after authors' response**
> >
> > Follow-up comments after authors' response.
> >
> > For Q1, I mentioned in the previous comments that the missing reference is Yeh et al., ICASSP 2018, not the CVPR 2017 paper.
> >
> > Yeh, Raymond A., et al. "Image Restoration with Deep Generative Models." 2018 IEEE International Conference on Acoustics, Speech and Signal Processing (ICASSP). IEEE, 2018.
> >
> > In Yeh et al. paper, the author did use the pre-trained generator to obtain the results. In addition, the ICASSP 2018 version did contain various image restoration tasks.
> >
> > For Q2, TV might not be the best solution for all these tasks, but it is a strong baseline. Reviewer #3 also mentioned that. The paper should include the results instead of just saying that it does not work. In addition, Yeh et al. paper should be the right baseline method to compare with.
> >
> > The rebuttal did not address my concerns. The score remains the same.

---

### Official Review · AnonReviewer3 · 2018-10-30
**The motivation is not convincing. The final model is too difficult to be optimized. The experimental results are also too weak for evaluation.**

**Rating:** 4
**Confidence:** 5

**Review:**

This paper proposed a framework to incorporate GAN into MAP inference process for general image restoration.

First, the motivation of the proposed framework is not convincing for me. That is, authors assumed that they have a degradation function F and all the inference process is just based on this known function. However, in real world scenarios, it is actually challenging to obtain exact degradation information. Thus we may only apply the proposed model on a few tasks with exactly known F.

Second, due to the norm based constraints, authors actually need to optimize a highly nonconvex optimization problem. Moreover, due to the trace based loss function, the computational cost will also be very high. Please notice that standard MAP based methods only need to solve a simple convex optimization model (e.g., TV) and these methods can also be applied for different restoration tasks. Actually, we only need to specify particular fidelity terms for different tasks. Moreover, very recent works have also successfully incorporate both generative and discriminative network architectures (e.g., [1,2]) into the optimization process. Therefore, I cannot find any advantage in the proposed method, compared with these existing MAP based image restoration approaches.

Finally, the experimental part is also too weak to evaluate the proposed method. As I have mentioned above, actually a lot of methods have been developed to address general image restoration tasks. Some works actually also incorporate generative and/or discriminative networks into MAP inference process for these tasks. Thus I believe authors must compare their method with these state-of-the-art approaches. Moreover, authors should conduct experiments on state-of-the-art benchmarks, including natural images. This is because the digitals images in MNIST do not have rich texture and detail structures, thus are not very challenging for standard image restoration methods.


[1]. Kai Zhang, Wangmeng Zuo, Shuhang Gu, Lei Zhang: Learning Deep CNN Denoiser Prior for Image Restoration. CVPR 2017: 2808-2817
[2]. Jiawei Zhang, Jin-shan Pan, Wei-Sheng Lai, Rynson W. H. Lau, Ming-Hsuan Yang: Learning Fully Convolutional Networks for Iterative Non-blind Deconvolution. CVPR 2017: 6969-6977

---

> ### Author Response · Authors · 2018-11-27
> **Response to AnonReviewer3**
>
> Thank you for your thoughtful review. We will address your concerns in turn.
>
> Q1: The degradation function F is challenging to obtain in real world scenarios.
>
> A1: Many state-of-the-art approach, including SRCNN and SRGAN, has their own implicitly defined degradation function. They use their function F to generate training samples during their training process, while we use our explicitly defined function F during the inference process. If the assumed degradation function F is not exactly the function in real scenarios, both these state-of-the-art approach and our method will suffer. So it is unfair to criticize our motivation just because we explicitly write out the degradation function F.
>
>
> Q2: TV can also be applied for different restoration tasks, and it is easier to be optimized.
>
> A2: We think you underestimate the difficulty of those restoration problems. Please check the degraded images in Table 3. These images are damaged so badly that TV cannot recover any meaningful thing. As a handcrafted prior, TV performs much worse than our data-driven baseline method in these tasks.

---

### Meta-Review · Area_Chair1 · 2018-12-11
**Lack of comparison with recent baselines**

**Confidence:** 5
**Recommendation:** Reject

**Metareview:**

This paper proposes a framework of image restoration by searching for a MAP in a trained GAN subject to a degradation constraint. Experiments on MNIST show good performance in restoring the images under different types of degradation.

The main problem as pointed out by R1 and R3 is that there has been rich literature of image restoration methods and also several recent works that also utilized GAN, but the authors failed to make comparison any of those baselines in the experiments. Additional experiments on natural images would provide more convincing evidence for the proposed algorithm.

The authors argue that the restoration tasks in the experiments are too difficult for TV to work. It would be great to provide actual experiments to verify the claim.